# GraphRAG-Bench: Challenging Domain-Specific Reasoning for Evaluating Graph Retrieval-Augmented Generation

## Abstract

Graph Retrieval Augmented Generation (GraphRAG) has garnered increasing recognition for its potential to enhance large language models (LLMs) by structurally organizing domain-specific corpora and facilitating complex reasoning. However, current evaluations of GraphRAG models predominantly rely on traditional question-answering datasets. Their limited scope in questions and evaluation metrics fails to comprehensively assess the reasoning capacity improvements enabled by GraphRAG models. To address this gap, we introduce GraphRAG-Bench, a large-scale, domain-specific benchmark designed to rigorously evaluate GraphRAG models. Our benchmark offers three key contributions: $(i)$ Challenging question design. Featuring college-level, domain-specific questions that demand multi-hop reasoning, the benchmark ensures that simple content retrieval is insufficient for problem-solving. For example, some questions require mathematical reasoning or programming. $(ii)$ Diverse task coverage. The dataset includes a broad spectrum of reasoning tasks, multiple-choice, true/false, multi-select, open-ended, and fill-in-the-blank. It spans 16 disciplines in twenty core textbooks. $(iii)$ Holistic evaluation framework. GraphRAG-Bench provides comprehensive assessment across the entire GraphRAG pipeline, including graph construction, knowledge retrieval, and answer generation. Beyond final-answer correctness, it evaluates the logical coherence of the reasoning process. By applying nine contemporary GraphRAG methods to GraphRAG-Bench, we demonstrate its utility in quantifying how graph-based structuring improves model reasoning capabilities. Our analysis reveals critical insights about graph architectures, retrieval efficacy, and reasoning capabilities, offering actionable guidance for the research community.

## 1 Introduction

Retrieval-Augmented Generation (RAG) [1; 2] has emerged as a key solution to ground large language models (LLMs) in external knowledge to mitigate both the hallucination problem and the lack of domain knowledge. By retrieving relevant text passages from corpora, RAG injects factual knowledge for a more reliable generation from LLMs. However, conventional RAG systems remain unsatisfactory when dealing with complex reasoning scenarios. The flat retrieval in RAG directly returns fragmentized chunks based on similarity matching, which limits their ability to model complex relationships between concepts to answer the questions requiring multi-hop reasoning [3; 4], i.e., 'What was the impact of [event] the 2008 Lehman Brothers bankruptcy on [person] Elon Musk's Tesla?' or global comprehension, i.e., 'What is the main idea of the [event] Trade Policy Change?'.

To address these limitations, Graph Retrieval-Augmented Generation (GraphRAG) has been extensively studied to capture the structured knowledge among concepts in the form of graphs [5; 6; 7],

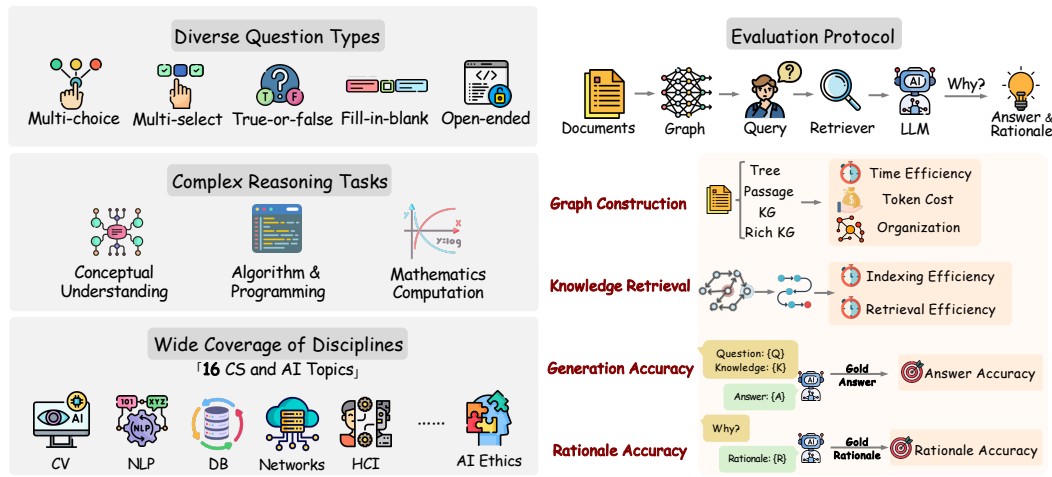

Figure 1: A sketched overview of our benchmark `GraphRAG-Bench`, illustrating the contributions.

where nodes represent concepts and edges are for the relations among them. Recent advances in GraphRAG can be categorized into three main directions. First, hierarchical graph construction methods like RAPTOR [8] and Microsoft's GraphRAG [5] organize knowledge through tree structures and community detection. Second, neural graph retrieval approaches, including GFM-RAG [9] and G-Retriever [10] employ graph neural encoders with specialized objectives for multi-hop reasoning. Third, dynamic knowledge integration systems such as DALK [11] and ToG [12] develop adaptive graph construction and traversal mechanisms that are tightly coupled with LLMs. By structuring knowledge as graphs, GraphRAG enables LLMs to both traverse and reason over explicit relational paths, but also supports deeper reasoning by inferring implicit relations based on the graph structure.

However, despite the promise, existing benchmarks for GraphRAG methods fail to reflect the performance of reasoning on graphs. They predominantly leverage the traditional QA dataset, e.g., HotpotQA [13], 2WikiMultiHopQA [14] and MuSiQue [15], which only feature explicit factoid questions with limited complexity and short answers, e.g., 'Who is the grandchild of Dambar Shah?'. These datasets suffer from three critical limitations: ($i$) There are only commonsense questions that could be probably covered in the training corpus of LLMs. ($ii$) They typically require only single-hop or shallow multi-hop reasoning based on explicit connections, which inadequately probes the unique advantages of graph-structured knowledge. ($iii$) Narrow Answer Formats. Most answers are short (names, dates) or multiple-choice, which could hardly reflect the reasoning ability over graphs. To this end, we would like to ask a research question:

*"Does graph augmentation truly enhance reasoning capabilities beyond simple retrieval?"*

In this paper, we propose GraphRAG-Bench, the first challenging domain-specific benchmark particularly designed for GraphRAG. ($i$) Our dataset contains 1,018 college-level question spans 16 disciplines, e.g., computer vision, computer networks, human-computer interaction, AI ethics, etc, featuring the ability of conceptual understanding, e.g., "Given [theorem] A and B, prove [conclusion] C", complex algorithmic programming, e.g., coding with interlinked function calls) and mathematical computation, e.g., "Given [Input], [Conv1], [MaxPool], [FC], calculate the output volume dimensions." ($ii$) GraphRAG-Bench contains five types of diverse questions to thoroughly evaluate different aspects of reasoning, including multiple-choice (MC), multi-select (MS), true-or-false (TF), fill-in-blank (FB) and open-ended (OE). ($iii$) We offer a comprehensive multi-dimensional evaluation on each component of GraphRAG, including graph construction, knowledge retrieval, answer generation and rationale generation. We aim to provide unprecedented insights into how graph-structured knowledge enhances LLMs' reasoning capabilities compared to traditional RAG approaches.

Overall, we propose the first challenging domain-specific benchmark, particularly concentrating on GraphRAG. It contains 1,018 questions in 5 question types spanning 16 topics and a corpus of 7 million words from 20 computer science textbooks. A comprehensive evaluation protocol is designed to stress-test GraphRAG methods on graph construction, retrieval, and multi-hop answer generation and rationale generation. Extensive experiments have been conducted with nine state-of-the-art

GraphRAG models. We make insightful observations and provide the insights that: 1) GraphRAG substantially enhances the reasoning capabilities of LLMs, and - to the best of our knowledge - we are the first to quantify this improvement using concrete evaluation metrics. 2) GraphRAG's impact varies by question types: it yields significant gains on some types but offers limited benefit for others.

## 2 Related Work

**GraphRAG.** Recent work in GraphRAG has focused on integrating structured knowledge and advanced retrieval strategies to overcome the limitations of vanilla RAG in handling large, noisy corpora and complex reasoning. For example, RAPTOR [8] and Microsoft's GraphRAG [5] both employ hierarchical clustering, RAPTOR via recursive tree construction with multi-level summarization, and GraphRAG via community detection with LLM-generated synopses, to support coarse-to-fine retrieval and diverse, high-coverage responses. GFM-RAG [9], G-Retriever [10], and LightRAG [16] each combine graph neural encoders with specialized retrieval objectives, respectively a query dependent GNN trained in two stages for multi-hop generalizability, a Prize Collecting Steiner Tree formulation to reduce hallucination and improve scalability, and a dual level graph augmented index for efficient, incrementally updatable lookup, to enable accurate, scalable reasoning over document graphs. Inspired by hippocampal memory processes, HippoRAG [17] leverages Personalized PageRank to achieve single-step multi-hop retrieval, delivering state-of-the-art efficiency and performance on both path following and path finding QA tasks. DALK [11] and KGP [18] introduce dynamic KG construction and traversal agents, using LLMs to build domain specific graphs and self aware retrieval policies, to inject structural context while reducing noise. ToG [12] tightly couples LLMs with KGs via beam search exploration, enabling iterative graph reasoning and on the fly correction without additional training. Collectively, these methods exemplify the GraphRAG paradigm by uniting graph structures, generative language models, and novel retrieval formulations to enhance knowledge integration, scalability, and deep reasoning across diverse domains.

**Prior benchmarks for GraphRAG.** To date, no dataset has been specifically designed for GraphRAG tasks. Widely used datasets such as Quality [19], PopQA [20], and HotpotQA [13] are tailored for general question answering, where answers can often be directly extracted from corpora, failing to effectively measure the core capabilities of GraphRAG methods. Multi-hop QA datasets like MusiqueQA [15] and 2WikiMultiHopQA [14] contain questions artificially constructed via rules and logic, rather than natural queries from real-world scenarios. Additionally, their corpora are short and often derived from converting entities and descriptions of existing KGs, which deviates from practical application contexts. While DIGIMON [7] benchmarks some methods, it neither introduces new datasets nor evaluates the reasoning capabilities of GraphRAG. Critically, all aforementioned datasets neglect question type distinctions, focusing primarily on simple questions and thus unable to reflect GraphRAG's performance variations across different question categories. In summary, existing datasets lack long contexts and raw documents, mismatching real-world scenarios, and omit gold rationale, making it impossible to systematically evaluate GraphRAG's reasoning abilities.

## 3 GraphRAG-Bench: Challenging Reasoning Benchmark for GraphRAG

### 3.1 Question design

To evaluate the GraphRAG framework on college-level reasoning, we first assembled an authoritative textbook corpus. Beginning with over 100 publications spanning 16 distinct subfields in computer science, we systematically identified the most representative 20 textbooks. We defined five types of questions, each targeting a different aspect of GraphRAG's reasoning capabilities, which are detailed in Tab. 1. After rigorous screening and refinement by several domain experts, we selected 1,018 high-quality challenging questions, covering a broad spectrum of topics.

By design, each question type is explicitly mapped to the core competencies of GraphRAG, with individual questions meticulously crafted for application in college-level instructional or assessment contexts. Should GraphRAG demonstrate improved performance on these tasks, it would establish itself as a highly effective tool in education, significantly enhancing teaching and learning efficiency.

| Question Type | Description |
|---|---|
| Fill-in-blank (FB) | Requires completing context-dependent statements with semantically precise terms. These assess the model's ability to generate contextually coherent content by leveraging local semantic dependencies and entity grounding within graph-structured knowledge. |
| Multi-choice (MC) | Presents a question with 4 options, including linguistically plausible distractors. These assess the model's capacity to discern correct answers through discriminative reasoning, integrating entity information and edge relationships to reject semantically similar but factually incorrect options. |
| Multi-select (MS) | Demands selecting 2–4 correct answers from 4 options, often requiring reasoning over interconnected concepts. The inclusion of overlapping distractors tests the model's ability to handle complex query semantics, aggregating evidence from multi-hop graph paths and resolving conflicts between related but non-essential attributes. |
| True-or-false (TF) | Involves verifying the correctness of statements. These measure the model's factual accuracy assessment, requiring logical inference over knowledge. |
| Open-ended (OE) | OE questions allow for a wide range of responses, requiring methods to formulate detailed and comprehensive answers. These evaluate the model's holistic knowledge synthesis, demanding the integration of multi-subfield knowledge to generate structured, logically coherent long-form responses. |

Table 1: The description of different question types.

## 3.2 Corpus collection and processing

Extracting accurate content from the 20 PDF-format core textbooks presents significant challenges. We implement a multi-stage pipeline comprising preprocessing, content parsing, post-processing, and hierarchy construction.

**Textbook Preprocessing.** 1) PDF Classification: To distinguish text-based pages from scanned (image-based) pages, we analyze each page's text density and image area proportion. Text-based pages are processed by extracting text directly using PyMuPDF, while scanned pages require optical character recognition (OCR) to extract their textual content. 2) Metadata Extraction: We extract metadata for each textbook, including its outline, total page count, and the page ranges for each chapter or section. This metadata supports the later construction of the document's logical structure.

**Content Parsing.** After preprocessing, we analyze each page's layout to extract textual and non-textual elements. 1) Layout Analysis: We apply LayoutLMv3 [21] for multimodal document layout analysis. LayoutLMv3 is pre-trained with masked language modeling, masked image modeling, and cross-modal alignment, enabling it to learn rich representations of document pages. The model classifies page regions into semantic categories such as titles, paragraphs, figures, tables, or decorative/irrelevant elements. This segmentation yields coherent content blocks on each page. 2) Formula Recognition: Mathematical formulas embedded in text are often misrecognized by OCR. To prevent this, we first detect inline formulas using a pre-trained YOLO-based model [22] from PDF-Extract-Kit. This model identifies the bounding boxes of formula regions so that formula images can be extracted separately, ensuring that OCR does not garble the formula content. 3) OCR: In scanned PDFs, OCR is applied to recognize text regions. We use PaddleOCR to transcribe text from the regions labeled as titles and body paragraphs via layout analysis. This step produces the page's textual content in the correct reading order, while preserving non-text elements as separate objects.

**Post-Processing.** After parsing, the extracted elements (text blocks, formula, figures, tables, etc.) may be disordered due to overlapping bounding boxes or fragmented text lines. We resolve these issues by reordering and merging page regions according to human reading order. Concretely, we use MinerU [23] for post-processing, which partitions each page into logical reading regions and sequences them so that the final text flow matches the natural reading sequence.

**Hierarchy Construction.** Finally, we organize the extracted content into a hierarchical textbook-tree structure. We map the textbook metadata (e.g., chapter titles, section divisions, and page ranges) to a

four-level hierarchy: Book Title → Chapter → Section (Subchapter) → Knowledge Content Unit. Each node in this hierarchy is annotated with its contextual metadata and its structural role. This textbook-tree provides an intuitive, pedagogical navigation framework aligned with the textbook's organization. The resulting corpus – with its accurate content extraction, structural annotation, and hierarchical organization – forms a robust basis for evaluating GraphRAG's ability to leverage organized textbook knowledge for context-rich reasoning and retrieval-augmented generation.

### 3.3 Expert-crafted rationale

Existing benchmarks typically supply only final answers or explicit graph paths; by contrast, our dataset supplies expert-crafted rationales that articulate the complete logical progression necessary to solve each problem. These rationales go beyond mere corpus aggregation; they are structured narratives that (i) isolate prerequisite concepts, (ii) describe the relationships among these concepts, and (iii) specify the inferential operations applied during problem solving. By tracing each step of logical inference and knowledge interaction, we can assess whether GraphRAG models truly generate contextually grounded explanations or simply exploit surface-level patterns.

To enable fine-grained, topic-specific evaluation, each question in our dataset carries two hierarchical labels: a broad subfield (Level 1, e.g., "Machine Learning") and a more granular concept (Level 2, e.g., "Unsupervised Learning"). These annotations structure our post-hoc analyses. For each topic, we measure not only the accuracy of the model's answer but also the degree to which its generated rationale aligns with the gold one. In this way, we convert evaluation into a multidimensional process, requiring models to produce both correct solutions and faithful reasoning patterns.

## 4 Experiments

We conduct experiments on each submodule following GraphRAG's pipeline, which includes the **graph construction** (or similar specialized structures), **knowledge retrieval**, and **generation**. Additionally, since our dataset contains a gold rationale for each query, we require the GraphRAG method to generate **rationales** during the generation phase to evaluate its reasoning capabilities.

**Metrics.** We provide a succinct introduction to the core ideas of each metric; the full evaluation protocol and details can be found in the Appendix.

- **Graph construction.** We evaluate graph construction across three aspects: 1) Efficiency: the time required to build a complete graph. 2) Cost: the number of tokens consumed during graph construction. 3) Organization: the proportion of non-isolated nodes within the constructed graph.

- **Knowledge retrieval.** We evaluate retrieval from two dimensions: 1) indexing time, defined as the duration required to construct the vector database for retrieval; 2) average retrieval time, representing the mean time consumed for knowledge retrieval per query. Additionally, we summarize the retrieval operators employed by each method to assess the complexity of their retrieval mechanisms.

- **Generation.** We argue that the existing exact match metric is inappropriate, as correct answering does not necessitate word-by-word correspondence. Therefore, this paper introduces a new metric, Accuracy, defined as follows: 1) For OE and FB questions, both the generated output and groundtruth are fed into an LLM via our designed prompt, which assigns a score based on semantic alignment and correctness. 2) For MC and TF, 1 point for the correct answer, 0 points for otherwise. 3) For MS, 1 point for a fully correct answer; 0.5 points for a subset; 0 points for incorrect answers.

- **Rationale.** We designed a prompt to feed both the rationale generated by the GraphRAG method and the gold rationale into a LLM, which assigns a reasoning score R to evaluate their semantic correspondence and reasoning consistency. Simultaneously, we developed an additional assessment metric, namely the AR metric, to determine whether the model is able to provide correct reasoning when it answers the question accurately. This metric serves to distinguish whether the model has merely guessed the correct answer or has actually engaged in proper logical reasoning to reach the correct answer, thereby offering a more comprehensive understanding of the model's performance.

**Experiment setups.** In our experiments, we evaluated the performance of nine state-of-the-art GraphRAG methods, including: 1) RAPTOR [8]; 2) LightRAG [16]; 3) GraphRAG [5]; 4) G-Retriever [10]; 5) HippoRAG [17]; 6) GFM-RAG [9]; 7) DALK [11]; 8) KGP [18]; 9) ToG [12]. To ensure a fair comparison across all methods, we adopted the same GPT-4o-mini as the default large

language model. We imposed no max token length to limit the performance of individual methods. For methods requiring top-k selection, we uniformly set k=5. Regarding text chunking, the chunk size was consistently set to 1200 tokens. Except for the parameters standardized for fair comparison, all other hyperparameters were configured to the optimal values reported in the original papers.

## 4.1 Evaluation of graph construction

| Method | Token cost of graph construction | Time cost of graph construction | Organization |
|---|---|---|---|
| RAPTOR (2024) | 10,142,221 | 20396.49s | - |
| KGP (2024) | 15,271,633 | 17318.07s | 46.03% |
| LightRAG (2024) | 83,909,073 | 12976.22s | 69.71% |
| GraphRAG (2025) | 79,929,698 | 11181.24s | 72.51% |
| G-Retriever (2024) | 32,948,161 | 5315.27s | 89.95% |
| HippoRAG (2024) | 33,006,198 | 5051.41s | 89.58% |
| DALK (2024) | 33,007,324 | **4674.30s** | 89.49% |
| ToG (2024) | 33,008,230 | 5235.30s | 89.95% |
| GFM-RAG (2025) | 32,766,094 | 5631.10s | **89.97%** |

Table 2: Comparison of graph construction process.

Graph construction aims to transform corpus into structured, storable objects, serving as the foundational step in GraphRAG. Current mainstream graph construction methods can be categorized into four classes: 1) Tree: RAPTOR leverages this structure, where each leaf node represents a chunk. By generating summaries via LLMs and applying clustering methods, parent nodes are iteratively created to form a hierarchical tree structure. 2) Passage Graph: Adopted by KGP, this structure represents each chunk as a node, with edges established through entity linking tools. 3) Knowledge Graph: Used in G-Retriever, HippoRAG, GFM-RAG, and DALK, this structure extracts entities and relationships from chunks using open information extraction (OpenIE) tools to construct knowledge graphs. 4) Rich Knowledge Graph: Employed by GraphRAG and LightRAG, this structure enriches standard knowledge graphs with additional information (e.g., summarizing descriptions for nodes or edges).

Experimental results in Tab. 2 show that the tree structure incurs the lowest token count, as it only invokes LLMs for summary generation, but requires the longest time due to iterative clustering. The passage graph has suboptimal token cost, invoking LLMs only for summarizing entities or relationships, with the second-longest time consumption attributed to the time-intensive entity linking process. The knowledge graph has moderate token usage, requiring LLMs for both entity extraction from corpora and triple generation from entities, yet achieves the shortest time consumption due to rapid knowledge graph construction after triple acquisition. The rich knowledge graph consumes the most tokens, as it generates additional descriptions for entities and relationships via LLMs on top of standard knowledge graphs, leading to increased time costs. For evaluating graph construction quality, we use the non-isolated nodes ratio as the metric. Since the Tree structure contains no isolated nodes, this metric is inapplicable to it. Experimental results show that the Knowledge Graph achieves the best performance, with its non-isolated nodes ratio maintained at approximately 90%. The Rich Knowledge Graph performs suboptimally; while it incorporates additional information, it inevitably introduces more noise. The Passage Graph exhibits the lowest non-isolated nodes ratio, indicating that entity linking tools fail to effectively establish edges between most entity pairs.

## 4.2 Evaluation of knowledge retrieval

As shown in Tab. 3. GFM-RAG incurs the shortest indexing time; it does not construct a traditional vector database to store entities but instead stores question-corresponding entities exclusively during graph construction. Among methods using vector databases, KGP, RAPTOR, and DALK exhibit lower costs due to minimal stored information; ToG, G-Retriever, and LightRAG have moderate costs, as relationship storage is inherently time-consuming; GraphRAG further increases indexing time by additionally storing community reports. HippoRAG demands the longest indexing time, attributed to its extra construction of entity<->relationship and relationship<->chunk mappings. Regarding average retrieval time, RAPTOR achieves the fastest speed, as its tree structure enables rapid information localization. GFM-RAG and HippoRAG follow, leveraging GNNs and PageRank algorithms for retrieval, respectively. G-retriever employs a prize-collecting Steiner forest algorithm, while LightRAG relies on relationship-based retrieval, both introducing additional latency. GraphRAG

needs to utilize community information for retrieval, which leads to its time-consuming. KGP, ToG, and DALK incur substantial time costs due to their dependence on LLM invocations during retrieval.

| Method | Retrieval operators | Indexing time | Average retrieval time |
|--------|---------------------|---------------|------------------------|
| KGP | Node | 204.10s | 89.38s |
| ToG | Node+Relationship | 1080.43s | 70.53s |
| GraphRAG | Node+Relationship+Chunk+Community | 1796.65s | 44.87s |
| DALK | Node+Subgraph | 407.10s | 26.80s |
| G-Retriever | Node+Relationship+Subgraph | 920.39s | 23.77s |
| LightRAG | Node+Relationship+Chunk | 1430.32s | 13.95s |
| HippoRAG | Node+Relationship+Chunk | 4695.29s | 2.44s |
| GFM-RAG | Node | **93.55s** | 1.96s |
| RAPTOR | Node | 451.03s | **0.02s** |

Table 3: Comparison of knowledge retrieval process.

### 4.3 Evaluation of generation accuray

| Method | Accuracy | | | | | |
|--------|----------|---|---|---|---|---|
| | Fill-in-blank | Multi-choice | Multi-select | True-or-false | Open-ended | Average |
| GPT-4o-mini | 74.29 | 81.11 | 76.68 | 75.95 | 52.23 | 70.68 |
| TF-IDF | 75.71 | 77.88 | 72.52 | 84.17 | 50.18 | 71.71↑ |
| BM-25 | 74.28 | 78.80 | 71.17 | **84.49** | 50.00 | 71.66↑ |
| DALK | 70.00 | 78.34 | 71.62 | 77.22 | 51.49 | 69.30↓ |
| G-Retriever | 70.95 | 77.42 | 71.62 | 78.80 | 52.04 | 69.84↓ |
| LightRAG | 65.24 | 78.80 | 73.42 | 82.59 | 53.16 | 71.22↑ |
| ToG | 70.48 | 78.80 | **78.38** | 79.75 | 54.28 | 71.71↑ |
| KGP | 74.29 | 79.26 | 74.77 | 82.28 | 51.49 | 71.86↑ |
| GFM-RAG | 72.38 | 80.65 | 72.07 | 82.59 | 52.79 | 72.10↑ |
| GraphRAG | 75.24 | **81.57** | 77.48 | 80.70 | 52.42 | 72.50↑ |
| HippoRAG | 70.48 | 80.18 | 74.32 | 81.65 | **56.13** | 72.64↑ |
| RAPTOR | **76.67** | 80.65 | 77.48 | 82.28 | 54.83 | **73.58**↑ |

Table 4: Comparison of generation process.

As shown in Tab.4. Given that GPT-4o-mini already exhibits strong question-answering capabilities, not all GraphRAG methods effectively enhance its performance. Notably, DALK and G-Retriever degrade LLM performance; their over-reliance on structural information at the expense of semantic content introduces excessive noise during generation, impairing LLM judgment accuracy. LightRAG, ToG, and KGP achieve slight performance improvements, indicating their retrieved content provides marginal assistance for generation tasks. In contrast, GFM-RAG, GraphRAG, and HippoRAG significantly boost LLM performance by effectively integrating graph structural information with chunk-level semantics: GFM-RAG leverages large-scale pretraining to obtain a robust foundation model, GraphRAG optimizes retrieval using community-based information, and HippoRAG enhances retrieval efficiency via PageRank algorithm. The top-performing method in experiments is RAPTOR, which constructs a tree structure through iterative clustering, a design that aligns with the natural hierarchical organization of textbook data, enabling efficient retrieval of relevant information. Additionally, most GraphRAG methods outperform traditional RAG baselines such as BM-25 and TF-IDF, highlighting the utility of graph-based architectures in improving generation accuray.

### 4.4 Evaluation of reasoning capabilities

As shown in Tab.5. In contrast to the high accuracy in generation tasks, GPT-4o-mini exhibits a notable decline in reasoning performance. The decrease in R score indicates that LLMs often fail to perform correct reasoning, instead selecting answers through conjecture or pattern matching in many cases. The drop in AR score suggests that even when LLMs provide correct answers, their reasoning processes may be flawed; alternatively, they might generate correct reasoning but choose incorrect answers. Importantly, all GraphRAG methods significantly enhance the reasoning capabilities of LLMs: through distinct algorithmic designs, these methods retrieve not only semantically relevant corpus for questions but also identify multi-hop dependent corpus in the knowledge base, providing evidential support for LLM reasoning. This enables LLMs to reason based on external information rather than relying solely on internal knowledge for conjecture. In terms of algorithm performance, the

distribution aligns with that of generation tasks: HippoRAG and RAPTOR remain the top performers, which is intuitive, since retrieving useful information is inherently correlated with enabling correct reasoning. Additionally, most GraphRAG methods still outperform traditional RAG baselines.

| Method | Reasoning | | | | | | | | | | | |
| | FB | | MC | | MS | | TF | | OE | | Average | |
| | R | AR | R | AR | R | AR | R | AR | R | AR | R | AR |
| GPT-4o-mini | 64.76 | 53.33 | 55.07 | 50.92 | 54.50 | 39.19 | 58.23 | 53.40 | 49.26 | 9.76 | 55.45 | 39.78 |
| TF-IDF | 68.09 | 52.61 | 52.76 | 49.19 | 56.30 | 43.02 | 64.08 | 61.23 | 50.37 | 10.50 | 57.61 | 42.38 |
| BM-25 | 69.04 | 56.42 | 57.14 | 53.11 | 57.20 | 42.79 | 65.18 | 62.18 | 50.74 | 11.52 | 59.18 | 44.15 |
| DALK | 70.95 | 55.24 | 54.15 | 50.35 | 59.01 | 46.40 | 62.18 | 58.23 | 54.09 | 9.67 | 58.89 | 42.12 |
| KGP | 64.29 | 49.29 | 56.45 | 52.07 | 58.11 | 44.37 | 64.08 | 60.68 | 52.42 | 8.92 | 58.74 | 42.22 |
| GraphRAG | **71.43** | 55.24 | 56.22 | 52.42 | 57.66 | 45.72 | 63.61 | 60.13 | 53.16 | 10.50 | 59.43 | 43.30 |
| G-Retriever | 70.00 | 55.00 | **57.60** | **53.46** | 60.81 | 48.20 | 64.24 | 60.21 | 53.35 | 10.04 | 60.17 | 43.66 |
| LightRAG | 66.19 | 47.86 | 57.14 | 52.30 | **61.71** | **49.10** | 66.61 | 63.45 | 53.16 | 10.13 | 60.46 | 43.81 |
| ToG | 70.00 | 53.10 | 56.00 | 51.73 | 57.21 | 45.72 | 65.66 | 62.26 | 54.46 | 12.08 | 60.17 | 44.01 |
| GFM-RAG | 70.00 | 54.76 | 56.22 | 52.07 | 58.11 | 45.50 | 66.46 | **63.69** | 53.72 | 10.69 | 60.36 | 44.30 |
| HippoRAG | 66.67 | 50.48 | 56.68 | 52.30 | 59.91 | 47.52 | **67.25** | 63.61 | **55.02** | 12.36 | **60.90** | 44.55 |
| RAPTOR | **71.43** | **57.86** | 56.45 | 52.07 | 60.36 | **49.10** | 66.30 | 62.90 | 53.90 | **13.57** | 60.81 | **45.53** |

Table 5: Comparison of reasoning ability.

## 4.5 Topic-specific generation accuracy analysis

Given our dataset spans 16 distinct topical domains, we conducted a fine-grained analysis of GraphRAG's impact on LLM generation accuracy. Overall, GraphRAG yields consistent improvements in most areas; However, several intriguing findings emerge: **1) Mathematics Domain.** All GraphRAG methods degrade the LLM's generation accuracy in mathematics. This is attributed to the critical reliance of mathematical problems on rigorous symbolic manipulation and precise reasoning chains; models must internally "compute" each deductive step rather than relying on keyword matching from external texts. Most documents retrieved through GraphRAG are explanatory or conceptual, with symbolic notation, formula layouts, and contextual structures often misaligned with the problem requirements, leading to ambiguities or loss of key steps during the extraction and transformation of information. **2) Ethics Domain.** Both GraphRAG and the LLM itself exhibit mediocre performance in ethics. We posit that ethical problems fundamentally involve subjective value judgments, whose meanings depend on dynamic contexts of moral trade-offs and social norms. The symbolic representations captured by LLMs through statistical learning struggle to accurately model ambiguous ethical constructs, introducing intrinsic limitations in reasoning. **3) Robustness.** Excellent GraphRAG approaches such as RAPTOR substantially enhance LLM generation accuracy across most topics, demonstrating robust performance that validates their cross-domain effectiveness.

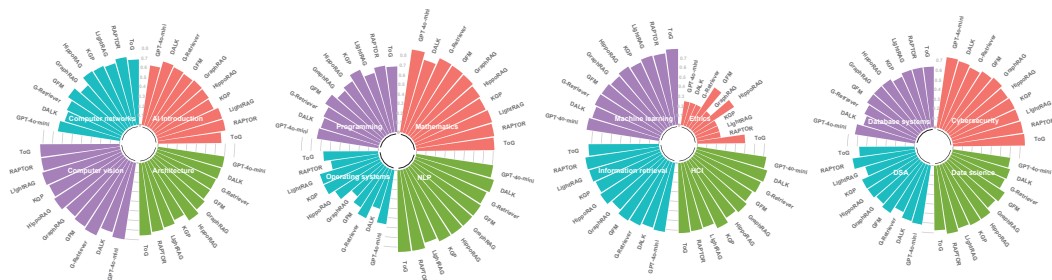

Figure 2: Comparison of Generation Accuracy by Topic.

## 4.6 Observation

*'Can GraphRAG improve performance across all question types?'*

**Accuracy drop of MC questions.** LLMs have internalized vast amounts of knowledge through extensive training on large corpora, enabling them to often correctly select answers in multiple-choice tasks. However, GraphRAG's retrieval-based augmentation may introduce redundant or loosely related information that does not precisely match the question context. Such retrieval noise can interfere with the model's decision-making ability, ultimately reducing its accuracy on MC questions.

**Improvement in TF questions.** TF questions require binary judgments about factual or logical statements. LLMs may contain blind spots or incomplete knowledge for certain facts, leading

to incorrect answers. By retrieving relevant factual evidence, GraphRAG helps the model verify statements before answering. These supplementals improve the model's accuracy on TF questions.

**Improvement in OE questions.** Open-ended questions allow for expansive, detailed responses, which can be challenging for LLMs that rely solely on their internal knowledge. GraphRAG mitigates this challenge by providing additional context and facts from external corpora. The retrieved information enriches the model's responses, improves subject-matter detail and expressiveness, and reduces instances of hallucination by grounding answers in explicit evidence.

**Different effects in FB & MS questions.** Fill-in-blank questions demand precise contextual understanding to correctly predict missing words. GraphRAG's retrieved corpora often fail to match exact contexts, introducing noise that degrades the model's performance on FB questions. Multi-select questions require choosing multiple correct answers from a set and involve reasoning over complex combinations of options; if GraphRAG's retrieval omits relevant answer options or includes irrelevant details, it can confuse the model. As a result, these question types place high demands on retrieval precision; GraphRAG may have limited benefit unless its retrieval is highly accurate.

*'Can GraphRAG effectively enhance LLMs' reasoning ability?'*

Experiments demonstrate that GraphRAG effectively enhances the reasoning capabilities of LLMs across diverse question types, increasing the probability of generating correct rationales alongside answers. This is attributed to their efficient retrieval mechanisms, which not only identify highly relevant corpora for questions but also provide robust evidential support for LLM reasoning processes. In particular, existing benchmarks lack systematic evaluation of GraphRAG's reasoning capabilities, an aspect of critical importance in real-world applications. For example, in the college-level educational context targeted in this document, users seeking professional knowledge expect not only correct answers, but also explicit rationales to facilitate understanding and knowledge acquisition. Similarly, in medical scenarios, patients require clear rationales for medication along with treatment recommendations to ensure transparency in decision-making. Thus, an effective GraphRAG approach should aim not only for high accuracy in answer generation but also for strong reasoning and explainability.

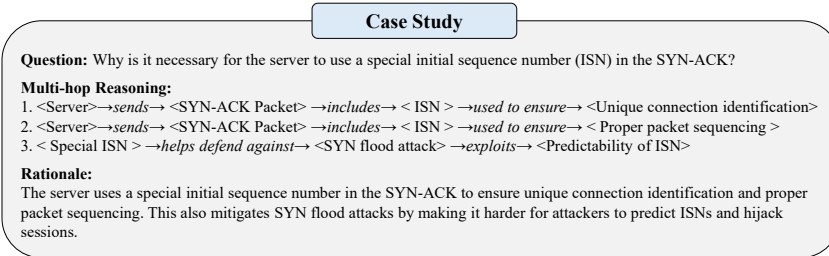

Figure 3: A case study in the topic of computer networks.

### 4.7 Case Study

As illustrated in Fig 3, we present a case study highlighting specific challenges within our dataset. Our questions span 16 core topics in undergraduate computer science; here, we focus on a sample from the Computer Networks section. This example demonstrates that (i) the questions demand specialized, college-level knowledge, and (ii) the correct answer cannot be retrieved through simple lookup. Instead, solving the problem requires synthesizing multiple reasoning steps to construct a coherent rationale before generating the final answer.

## 5 Conclusion

In this paper, we present GraphRAG-Bench, the first domain-specific benchmark designed for GraphRAG, comprising a 16-discipline dataset that challenges methods with multi-hop reasoning, complex algorithmic/programming tasks, mathematical computing, and varied question types. Our comprehensive, multi-dimensional evaluation, spanning graph construction, knowledge retrieval, generation and reasoning, quantifies the enhancement of LLM reasoning when augmented with structured knowledge. Extensive experiments on nine state-of-the-art GraphRAG methods reveal the significant role of graph integration in improving reasoning and generation performance.

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
