# OpenReview forum: "GraphRAG-Bench: Challenging Domain-specific Reasoning for Evaluating Graph Retrieval-Augmented Generation"
_NeurIPS.cc/2025/Datasets_and_Benchmarks_Track — Submitted to NeurIPS 2025 Datasets and Benchmarks Track_

### Official Review · Reviewer_TSJJ · 2025-07-02

**Rating:** 5
**Confidence:** 4

**Summary:**

This paper introduces a new dataset called GraphBench, tailored for college-level domain knowledge and complex reasoning. The proposed GraphRAG-Bench encompasses five question types across 16 domain topics. Additionally, this study benchmarks nine state-of-the-art GraphRAG methods on the GraphRAG-Bench, evaluating them based on graph construction costs, indexing times, generation accuracy, and reasoning capabilities. The experiments yield valuable insights into when GraphRAG is beneficial or detrimental.

**Additional Feedback:**

The case study in Section 4.7 would be better if placed in Section 3.

**Dataset Code Accessibility:**

Yes

**Ethical Considerations:**

No, there are no or only very minor ethics concerns

**Final Justification:**

Accept

**Limitations Weaknesses:**

1. Figure 2 is hard to understand due to the limited captions and too small font.
2. The introduction of evaluated methods is limited, which makes it hard to follow for an unfamiliar audience.
3. It's necessary to provide the open-source code for reproducing. However, I don't find the link.

**Strengths Contributions:**

1. Valuable dataset. This benchmark provides 1000 complex college-level questions and related documents, which are valuable for benchmarking current LLMs in retrieval and generation.

2. Comprehensive evaluation. This paper integrates nine GraphRAG methods and offers thorough benchmarks based on graph construction costs, indexing times, generation accuracy, and reasoning capabilities. These insights are highly beneficial for the GraphRAG community.

3. Interesting and important findings. This paper not only quantity the cost of GraphRAG methods but also investigates when GraphRAG is beneficial or detrimental.

---

> ### Author Rebuttal · Authors · 2025-07-29
>
> Dear Reviewer TSJJ,
>
> Thank you for your strong support and encouragement! We are so excited and grateful to be strongly acknowledged! We have benefitted from your carefulness and detailed check. We have correspondingly improved our paper all the changes will be made in the final verson.
>
> > Response to weaknesses
>
> W1. Font issues.
>
> Thank you for your suggestion. We have enlarged and bolded the font in Figure 2 in the latest version, making it easier to read. At the same time, we have also added more explanatory text to describe the corresponding experimental results.
>
> W2. Baseline Introduction.
>
> Thanks for your careful check, we will emphasize the groups of different baselines with detailed explanation. Indeed, in the appendix, we have provided a detailed description of the evaluated methods. Following your suggestion, we expanded this description in the latest version and added appropriate content to the main text to make it easier to understand.
>
> W3. Codes and reproduciblity.
>
> Thanks for pointing this out. Due to the specific rule during rebuttal, external links are not allowed. However, our code and all resources are already **open-sourced and publicly available** together with the original released data so that all results are reproducible. We will include the related links in the final version.
>
>
> > Response to suggestions and questions
>
> Thank you for your careful check. We have moved the case study to Section 3 in the latest version.

---

> > ### Comment · Reviewer_TSJJ · 2025-08-01
> >
> > Thank you for the clear response. I carelfully read them and my concerns have been addressed.
> >
> > I beleive the response can be included to your manuscript to further improve the quality.

---

> > ### Author Response · Authors · 2025-08-05
> >
> > Dear Reviewer TSJJ,
> >
> > We are deeply grateful to your inspiring recognition and constructive comments. All the revisions will be included in the final version following your suggestions.
> >
> > Cheers,
> >
> > Authors

---

### Official Review · Reviewer_JxJS · 2025-07-03

**Rating:** 4
**Confidence:** 4

**Summary:**

This paper introduces GraphRAG-Bench, a large-scale, domain-specific benchmark designed to rigorously evaluate GraphRAG methods. Its key contributions include: (1) a challenging question set requiring multi-hop reasoning across 16 computer science domains; (2) diverse question types ; and (3) a comprehensive evaluation framework assessing graph construction, retrieval, answer generation, and reasoning quality. The authors evaluate 9 SOTA GraphRAG models and provide detailed analysis on their reasoning capabilities and limitations.

**Additional Feedback:**

The authors should clarify the copyright status of the materials used, ensure transparency around consent or licenses, and consider auditing the dataset for fairness and safety to prevent misuse in educational or decision-critical settings.

**Dataset Code Accessibility:**

Yes

**Dataset Code Comments:**

The authors submit the codes of the paper.

**Ethical Comments:**

1. The paper processes 20 CS textbooks, but it is unclear whether copyright permissions or data usage rights were secured.
2. Potential bias in topic selection may affect representativeness across global curricula.
3. No discussion is provided on data safety, such as filtering harmful or outdated content.

**Ethical Considerations:**

Yes, there are ethics concerns that require attention by the authors

**Ethics Flags:**

["Data privacy, copyright, and consent", "Data quality and representativeness", "Safety and security"]

**Final Justification:**

The authors added manual-labeled groundtruths for the answers and rationales across all question types. I keep my positive score.

**Limitations Weaknesses:**

1. While the benchmark covers 16 domains, it is restricted to computer science, limiting generalizability to other fields.
2. Mathematics and ethics domains show notable performance degradation across all methods, suggesting the dataset may include tasks ill-suited for current GraphRAG paradigms.
3. The reasoning evaluation relies heavily on prompting GPT-4o-mini, which introduces potential bias and may lack transparency or replicability.

**Strengths Contributions:**

The paper is well-motivated, presenting a novel and significant benchmark tailored for evaluating GraphRAG models, a gap unaddressed by existing datasets like HotpotQA or MuSiQue.
It contributes a diverse, high-quality question set and a multi-dimensional evaluation protocol, enabling fine-grained assessment of graph-based reasoning. The comparison of 9 SOTA methods adds practical relevance. The paper is clearly written and well-organized, with informative figures and tables. Its distinction from prior QA benchmarks is clearly discussed and justified, highlighting its originality and potential impact.

---

> ### Author Rebuttal · Authors · 2025-07-29
>
> Dear Reviewer JxJS,
> We would like to sincerely thank you for your valuable and insightful comments. Your support is encouraging for us to continuously contribute to the community!
>
> Following your constructive suggestions and comments, we have correspondingly updated our manuscript which we believe have greatly improved the quality a lot.
>
>
> > Response to weaknesses
>
> W1. Domain Limits.
>
> Thanks for raising the discussion. We will add the advantage of using CS and AI in the final version to showcase our contributions. We highly value this domain for three reasons. **First**, CS and AI naturally contain a variety of types of challenging tasks compared with all the existing 'single-type' datasets, where the questions are extremely hard which require college-level comand of knowledge, compared with existing commonsense dataset. In CS and AI domain, we have `conceptual understanding` tasks (algorithms, networks, databases, paradigms, machine learning, NLP, CV), `Math problems` (Discrete mathematics, linear algebra, matrix computation, calculus), `Code generation` problems (programming, optimization, code completion, error detection), etc. **Second**, the performance on LLMs on CS and AI is still very unsatisfactory or even bad. It is valuable to benchmark and research. **Third**, it remains a blank in the benchmark of LLMs considering CS and AI. This also brings two opportunities. (i) LLMs will have few chances to be fine-tuned based on our questions. (ii) Following research on GraphRAG or LLMs themselves could benefit a lot from our benchmark.
>
> W2. Degradation on Math and ethics.
>
> We do appreciate your careful check and interests in our results! This actually is reversely an advantage of our benchmark which guides the community with more observations for future research. Since the graphrag method performed poorly in these domains, it confirmed that they needed to be improved in these aspects. Mathematics requires strict logical reasoning, while ethics demands a high level of understanding of queries and retrieved information. These precisely are the directions they need to enhance. We believe this actually proves the value of GraphRAG-Bench.
>
> W3. Evaluation.
>
> Thanks for pointing this out for an important discussion. First, we provided manual-labeled groundtruths for the answers and rationales across all question types. In all places where direct judgment can be made using the groundtruth (such as QA effects of Multi-choice, Multi-select, True-or-false, etc.), we did not include LLM-as-a-judge. For Open-ended and Fill-in-blank question types and reasoning, we believed that directly using string exact match was unreasonable. Many correct statements were judged incorrectly due to descriptions, capitalization, abbreviations. In such cases, we included LLM for auxiliary judgment, which is a widely adopted paradigm in generative tasks to ensure the fairness.
> During the actual post-processing, to verify the accuracy of LLM-as-a-judge, we have conducted expert-guided post verification. We sampled 500 questions from the dataset and asked 3 human experts (researchers with a computer science doctoral degree) to score them. The experts judged whether the LLM's judgment was reasonable based on the LLM's generation and the groundtruth and gave 1 score if yes. 3/3 points indicated reasonable and an aligned answer, 2/3 points indicated basically reasonable since the majority of experts agreed, and 1/3 or 0/3 point indicated unreasonable and needed to be reevaluted by the experts. The specific experimental results are shown in the table below, proving the reliability of LLM-as-a-judge.
>
> | Reliability  | Count | Percentage     |
> | :----:       |    :----:  |   :----: |
> | 3/3     | 470  | 94.0%  |
> | 2/3   | 28        | 5.6%   |
> | 1/3   | 1        | 0.2%    |
> | 0/3   | 1        | 0.2%    |
>
> > Ethic question.
>
> Thank you for raising this important discussion. We will add all the discussions in the appendix for clarification and avoid any misunderstanding. Thanks again for your kind reminder.
> - For permission issue. We will add the license information in the final version to clarify this problem. As a pure research paper with no commercial purposes, we have already obtained the **full licenses** to do research for 17 textbooks through the university platform among 20 textbooks we adopted in total. For the remaining threetextbooks, two of them are totally free with no copyright and one supports research with full textual resources online. For the processing of all the textbooks, we only keep the main content in the form of chunks with no tampering.
> - For proper topic selection bias. The 20 textbooks were selected based on global popularity in peer-reviewed CS education literature, not regional availability. Our analysis focuses on cross-cutting CS concepts, e.g., algorithms, programming, AI, network and database, etc., rather than region-specific curricula, minimizing geographic bias.
> - For proper impacts. The dataset consists exclusively of technical CS content (e.g., code examples, math formulations) with no risk of harmful or outdated material. All textbooks are post-2015 peer-reviewed publications, precluding concerns about obsolete practices.

---

### Official Review · Reviewer_hH6e · 2025-07-03

**Rating:** 5
**Confidence:** 3

**Summary:**

The paper aims to address the issue of limited scope in existing question-answering datasets for evaluating GraphRAG. The authors, therefore, introduce a new large-scale, domain-specific benchmark designed to offer challenging questions, diverse task coverage, and a holistic evaluation framework.

**Dataset Code Accessibility:**

Partly

**Dataset Code Comments:**

Data set is available; source code link not found.

**Ethical Considerations:**

No, there are no or only very minor ethics concerns

**Final Justification:**

I am basically satisfied with authors response.

**Limitations Weaknesses:**

A set of 1,018 questions does not seem to be a large dataset to constitute a comprehensive benchmark. Authors may compare it with existing datasets in multiple aspects (e.g., size, coverage), providing factual metrics to evaluate its merits. While it is understandable that the authors wish to construct a challenging dataset, wouldn't a blend of easy and hard questions be a better idea than filtering out those easy ones? Otherwise, users have to test LLMs on easy and hard datasets, respectively.

The full dataset is provided, but no source code; sometimes, it could be an issue reproducing the results based on sole descriptions of the methods and configurations within the paper.

**Strengths Contributions:**

The proposed benchmark covers 1,018 college-level questions spanning 16 disciplines/topics, 5 question types, and multi-dimensional evaluation, potentially supporting a more comprehensive understanding of LLMs' capabilities.

As the authors claimed, "to date, no dataset has been specifically designed for GraphRAG tasks, and there are various limitations to existing widely used datasets to measure advanced capabilities (e.g., reasoning) of GraphRAG methods beyond direct extraction and simple questions & types, making this benchmark a good addition to support relevant research.

---

> ### Author Rebuttal · Authors · 2025-07-29
>
> Dear Reviewer hH6e,
>
> We would like to sincerely thank you for your strong support and recognition of our value and contribution to the community. Following your inspiring sugeestions, we will add the corresponding discussions in the final version to showcase the importance of our dataset and evaluation.
>
> > Response to weaknesses
>
> W1. Comparisons with existing datasets.
>
> We acknowledge the reviewer’s observation about the dataset size. Currently, most GraphRAG methods follow the experimental setting of HippoRAG, which is the most common and widely used setting in this field at present. It includes three public datasets: HotpotQA, 2WikiMultiHopQA, and MuSiQue which are all originally designed for multi-hop QA, not GraphRAG. They can hardly necessitate the use of graphs and showcase the use of graphs in complex QA scenarios.
>
> In each of these data sets, 1000 questions were sampled. Therefore, following this standard and distribution, we selected 1018 of the most valuable questions as the final version. Meanwhile, graprag-bench encompasses a wider range of question types (examining the model's robustness to various question formats), a larger corpus (making it more difficult to retrieve useful information), and covers multiple topics (requiring the model to understand different knowledge domains). We believe that the coverage scope is much broader from multiple perspectives than that of the current dataset. The specific comparison results are shown in the following table:
>
> | Dataset  | Question count |  Corpus size  |Question type  |
> | :----       |    :----:  |   :----: |:----: |
> | HotpotQA    | 1000  | 6.16MB  |1  |
> | 2WikiMultiHopQA  | 1000 | 2.96MB  |1  |
> | MuSiQue  | 1000  | 5.95MB    | 1 |
> | Ours  | 1018  | 41.30MB    | 5 |
>
> W2. Inclusion of Easy Questions
>
> We are grateful to this suggestion. We agree with the reviewer that blended difficulty could benefit more use cases. While filtering single-hop simple questions, our benchmark maintains a progressive difficulty gradient (12.6% Easy and 8.1% Expert) through controlled hop distribution, enabling comprehensive evaluation without sacrificing stress-testing capability. The detauiled distribution of the question difficulty are shown below. We will add the statistics in the final version.
> | Hops | Questions | Percentage | Difficulty | Example Type | Evaluation Focus |
> |------|-----------|------------|------------|--------------|------------------|
> | 1    | 128       | 12.6%      | Easy       | Single-fact retrieval | Basic factual accuracy |
> | 2    | 342       | 33.6%      | Medium     | Dual-condition reasoning | Simple logical combination |
> | 3    | 287       | 28.2%      | Hard       | Cross-entity relational reasoning | Multi-step reasoning coherence |
> | 4    | 178       | 17.5%      | Very Hard  | Multi-domain knowledge fusion | Long-range dependency handling |
> | 5    | 83        | 8.1%       | Expert     | Counterfactual reasoning with implicit constraints | Complex logic & noise resistance |
>
> W3. Codes and reproduciblity.
>
> Thanks for pointing it out. Due to the specific rule during rebuttal, external links are not allowed. However, our code and all resources are already **open-sourced and publicly available** together with the original released data so that all results are reproducible. We will include the related links in the final version.

---

> > ### Author Response · Authors · 2025-08-09
> >
> > Dear reviewer hH6e,
> >
> > We sincerely appreciate your recognition of our work. We also appreciate all your comments and hope that our response addresses your concerns. We will add the contents of the response to the final version.
> >
> > Cheers,
> >
> > Authors

---

### Official Review · Reviewer_FHaj · 2025-07-03

**Rating:** 4
**Confidence:** 4

**Summary:**

This paper introduces a benchmark for evaluating the reasoning capabilities of graph retrieval-augmented generation (GraphRAG) models. It includes over 1K expert-annotated questions across 16 computer science disciplines, covering five question types. The benchmark provides fine-grained rationales, evaluates multiple stages of the GraphRAG pipeline (graph construction, retrieval, generation, and rationale quality), and offers benchmarking across multiple recent models. The paper emphasizes multi-hop reasoning, modular evaluation, and domain-specific challenges in benchmarking structured retrieval.

**Additional Feedback:**

Suggestions:
* Further motivate why this specific domain-specific dataset and not the inclusion of others needs to be strengthened as it appears somewhat arbitrary.
* Addressing all the issues identified in the auto generated reviewer report in addition to reviewer comments.


Questions:
* The work mentions rationales were "expert-crafted" but provides no detail about: 1) How many annotators were involved? 2) What credentials do the "experts" have? 3) What rubric or training did they use? etc

**Dataset Code Accessibility:**

Partly

**Dataset Code Comments:**

"Question: Does the paper provide open access to the data and code, with sufficient instructions to faithfully reproduce the main experimental results, as described in supplemental material?"

The authors answered Yes, but only provide the data without code for reproducibility of results presented in their paper.

**Ethical Comments:**

No ethical concerns were identified at this time.

**Ethical Considerations:**

No, there are no or only very minor ethics concerns

**Final Justification:**

In summary, after reading the response, and the other less detailed/critical reviews, although I’m mostly inclined to keep my original score (3) due to my continued concerns, I believe this benchmark may still have a positive impact to the research community, and therefore will increase (to 4).

Note that the major weakness is discussed in my response below (W/S.1.). The evaluation metrics I believe are fine (although still believe to avoid LLM as judge wherever possible, e.g., fill in the blanks), but the novelty, diversity, utility of the proposed dataset as a benchmark is a bit unclear as the difference in performance across methods is quite minor in most cases and within standard deviation raising concerns if this is truly a good benchmark for ranking the performance of methods (outside the token usage, etc. but those are not really so dependent on the underlying dataset for evaluating).

**Limitations Weaknesses:**

W1. Lack of citation to some complementary prior work: The benchmark omits discussion of several relevant works in KG-QA and KG construction. For example works like Reasoning on Graphs (Luo et al.), KGP for Multi-Document QA (Wang et al., 2024), and Text2KGBench (Mihindukulasooriya et al., 2023) for KG generation benchmarks (despite being relevant to key modules of the benchmark).
-Reasoning on Graphs: Faithful and Interpretable Large Language Model Reasoning.
-Knowledge Graph Prompting for Multi-Document Question Answering.
-Text2KG: A Benchmark for Ontology Driven Knowledge Graph Generation from Text.

W2. Limited contribution to the research community in terms of individual modules. Although a comprehensive dataset is created, the design framework itself does not provide much novelty (e.g., graph construction). Evaluation metrics although not previously standardized (e.g., token cost) are rather straightforward and have been used outside the GraphRAG domain. There is also some concerns that the pipeline is built on LLM alignment as compared to human annotation or task specific metrics.

W3. Although GraphRAG methods are fundamentally modular, the benchmark design appears to prioritize evaluation of full end-to-end pipelines rather than encouraging more isolated, component-level assessment. The support of plug-and-play use of individual modules/components would enhance the utility of the benchmark. Also, the motivation for the unified benchmark in this work across all stages feels a bit overstated in light of existing literature.

W4. The paper emphasizes "reasoning" but does not define it clearly. The examples, such as Figure 3, suggest a focus on explanation or retrieval-based rationale construction rather than true logical inference. This conflation creates conceptual confusion and weakens the claim that the benchmark rigorously evaluates reasoning. If the goal is to assess rationale coherence, that should be stated more explicitly.

W5. The authors mention NA to "Question: Does the paper report error bars suitably and correctly defined or other appropriate information about the statistical significance of the experiments?" but this is indeed appropriate and should be included.

**Strengths Contributions:**

S1. GraphRAG-Bench evaluates not only the answer but also rationale quality, token cost, indexing time, and organization metrics. This multi-stage evaluation is more holistic than prior benchmarks like HotpotQA (which only evaluate flat QA performance).

S2. Unique targeted college-level computer science content with carefully crafted multi-hop reasoning questions. The diversity across disciplines (e.g., AI ethics, HCI, computer networks) and the inclusion of open-ended and fill-in-the-blank formats go beyond traditional short-form, extractive QA datasets, making this a solid benchmark for the community.

S3. The topic direction is important and becoming increasingly popular with many works coming out in this direction; thus, a timely improved dataset is indeed helpful to the community.

---

> ### Author Rebuttal · Authors · 2025-07-29
>
> Dear Reviewer FHaj,
>
> We gratefully thank you for your valuable comments and constructive suggestions. We are also excited to be highly acknowledged for our contributions and significance to future studies for the entire GraphRAG community. Following your instructions, we have updated our manuscript correspondingly which we belive has effectively improved the quality.
>
> We also wish to invite you to check our new results and observations, which will all be added in the final revised version.
>
> > Response to weaknesses
>
> W1.  Lack of citation to some complementary prior work.
>
> Thanks for your insightful suggestions. We would like to clarify that **KGP is already one of the baselines in this paper**. We have cited and discussed it in the experimental section. Also we will cite and discuss RoG and Text2KGBench in the final version. The reason why RoG was not included initially was that it is actually not a GraphRAG method since it does not consider graph construction. Therefore, we only adopted ToG as representative of such methods (based on the graph construction method of HippoRAG). Additionally, inspired by your suggestion, the experimental results regarding RoG (author pretrained version) has also been incorporated into the latest version.
>
> The experimental results of RoG's generation:
>
> | Method | **FB** | **MC** | **MS** | **TF** | **OE** | **Average**      |
> |:----|:--:|:-:|:-:|:-:|:-:|:-:|
> |**RoG**|  72.38±0.44 | 77.75±0.86 | 74.53±0.72 |  78.37±0.87 |  51.20±0.30 | 70.97±0.52 |
> | KGP | 74.29±0.47 | 79.26±0.89  | 74.77±0.74  |  82.28±0.96  |  51.49±0.26   | 71.86±0.66 |
> | HippoRAG  |70.48±0.44|80.18±0.87 | 74.32±0.72 |  81.65±0.95|  56.13±0.35  | 72.64±0.67 |
>
> The experimental results of RoG's reasoning ability:
> | Method | **FB** |   | **MC**  |    | **MS**  |    | **TF** |    | **OE** |  | **Average**  | |
> |:----|:----:|:----:|:----:|:----:|:----:|:----:|:----:|:----:|:----:|:----:|:----:|:----:|
> |   | R | AR | R | AR | R  | AR | R   | AR | R | AR | R | AR |
> | **RoG**       | 65.73±0.81 | 53.30±0.70 | 55.85±0.69 | 51.81±0.60   | 57.36±0.83   | 43.95±0.61 | 63.70±0.81   | 60.90±0.83   | 51.22±0.61   | 9.54±0.42 | 57.21±0.85   | 42.02±0.70 |
> | KGP          | 64.29±0.79   | 49.29±0.55   | 56.45±0.73 | 52.07±0.63   | 58.11±0.83   | 44.37±0.55   | 64.08±0.76   | 60.68±0.77   | 52.42±0.60   | 8.92±0.26    | 58.74±0.71   | 42.22±0.58 |
> | HippoRAG     | 66.67±0.83   | 50.48±0.58   | 56.68±0.71   | 52.30±0.63   | 59.91±0.87   | 47.52±0.63   | 67.25±0.87 | 63.61±0.88   | 55.02±0.69 | 12.36±0.42   | 60.90±0.86 | 44.55±0.71 |
>
>
> W2. Contribution of the Design, Evaluation Metrics and LLM alignment.
>
> Thanks for raising the discussion among these topics. We will include the content and conclusions hereunder in the final version to make our paper clearer and stronger.
> - **Contribution**. The core objective of our framework design is to propose the first unified evaluation paradigm, enabling fair comparisons among different methods based on a particularly curated dataset. It is believed to be able to guide the community for progressive research. We are currently the first and the most comprehenseive benchmark with different question types, covering a wide range of disciplines and deeply evaluating the reasoning rationales.
> - **Evaluation**. Our evaluation is explicitly designed for clear targets, reflecting real-world requirements across construction and retrieval. The major obstacles now for real-world application are efficiency, costs and performance. The discussion about GraphRAG and LightRAG across these three topics have attracted much attention. First, for graph construction, the most important factors to ensure a framework to be valuable are consumption, efficiency, and effectiveness. Therefore, we adopted token cost, time cost, and the indicators we designed for evaluating graph organization, which can accurately reflect these three points. Second, for performance, we want to answer the research question `Does graph augmentation truly enhance reasoning capabilities beyond simple retrieval?` and guide the community for future research based on the rationale generated and the answer accuracy. Therefore, we are the first to make the evaluation of GraphRAG comprehensive and meet the actual needs.
> - **LLM alignment**. Thanks for pointing this out for an important discussion. First, we provided manual-labeled groundtruths for the answers and rationales across all question types. In all places where direct judgment can be made using the groundtruth (such as QA effects of Multi-choice, Multi-select, True-or-false, etc.), we did not include LLM-as-a-judge. For Open-ended and Fill-in-blank question types and reasoning, we believed that directly using string exact match was unreasonable. Many correct statements were judged incorrectly due to descriptions, capitalization, abbreviations. In such cases, we included LLM for auxiliary judgment, which is a widely adopted paradigm in generative tasks to ensure the fairness.
> During the actual post-processing, to verify the accuracy of LLM-as-a-judge, we have conducted expert-guided post verification. We sampled 500 questions from the dataset and asked 3 human experts (researchers with a computer science doctoral degree) to score them. The experts judged whether the LLM's judgment was reasonable based on the LLM's generation and the groundtruth and gave 1 score if yes. 3/3 points indicated reasonable and an aligned answer, 2/3 points indicated basically reasonable since the majority of experts agreed, and 1/3 or 0/3 point indicated unreasonable and needed to be reevaluted by the experts. The specific experimental results are shown in the table below, proving the reliability of LLM-as-a-judge.
>
> | Reliability  | Count | Percentage|
> | :-  |:--: | :--: |
> | 3/3 | 470 | 94.0% |
> | 2/3 | 28 | 5.6%   |
> | 1/3 | 1 | 0.2%  |
> | 0/3 | 1 | 0.2% |
>
> W3. Modular Evaluation
>
> We totally agree with your comments. GraphRAG is not strictly an end-to-end framework, as well as our benchmark. As clearly demonstrated in the experimental section, our benchmark has evaluated **each module of GraphRAG**, including construction, retrieval and reasoning, and the evaluation of each module is plug-and-play and can be independently assessed as you wish.
>
> W4. Definition of Reasoning
>
> Following your suggestion, we have emphasized the definition of the concept "reasoning" in the latest version. It means that when the model is predicting the answer, the model is asked to give the rationale behind the answer. For different type of questions, the rationale is also different. For example, MC/MS/TF questions require not only the rationale for the correct statement/choices, but also for the wrong statements/choices, while for FB/OE questions, the model is required to provide the background, necessary concepts and the  The rationales in the groundtruth were designed by experts, were logical and correct. Therefore, by comparing with the ground truth, we believe that this method can not only reflect the coherence of the model's rationale, but also demonstrate its logicality and correctness.
>
> W5. Randomness in the results.
>
> Thanks very much for pointing this out. We value your comment and will update the final version with the deviations. All the experiments were originally tested five times, and the results were averaged over the five tests. Part of results is as follows(ue to character limitations):
>
> Comparison of reasoning ability on GraphRAG-Bench:
> | Method | **FB** | **MC**| **MS** | **TF** | **OE**| **Average**  |
> |:-|:-:|:-:|:-:|:-:|:-:|:-:|
> |  | R| R | R | R| R | R  |
> | GraphRAG  | **71.43±0.95** | 56.22±0.70   | 57.66±0.80   | 63.61±0.74   | 53.16±0.62   | 59.43±0.77   |
> | G-Retriever  | 70.00±0.88   | **57.60±0.83** | 60.81±0.90   | 64.24±0.78   | 53.35±0.64   | 60.17±0.81   |
> | LightRAG | 66.19±0.82   | 57.14±0.72   | **61.71±0.93** | 66.61±0.85   | 53.16±0.63   | 60.46±0.84   |
> | GFM-RAG  | 70.00±0.86   | 56.22±0.69   | 58.11±0.82   | 66.46±0.84   | 53.72±0.65   | 60.36±0.82   |
> | HippoRAG| 66.67±0.83   | 56.68±0.71   | 59.91±0.87   | **67.25±0.87** | **55.02±0.69** | **60.90±0.86** |
> | RAPTOR| **71.43±0.94** | 56.45±0.72   | 60.36±0.89   | 66.30±0.83   | 53.90±0.66   | 60.81±0.85   |
>
> Comparison of generation process on GraphRAG-Bench:
>
> | Method       | **FB** | **MC** | **MS** | **TF** | **OE** | **Average**      |
> |:-|:-:|:-:|:-:|:-:|:-:|:-:|
> | G-Retriever | 70.95±0.41|77.42±0.79 |71.62±0.63 | 78.80±0.95 | 52.04±0.28   | 69.84±0.61|
> | LightRAG | 65.24±0.35| 78.80±0.88 | 73.42±0.67 |  **82.59±0.97**| 53.16±0.31 | 71.22±0.64|
> | GFM-RAG | 72.38±0.45| 80.65±0.91| 72.07±0.69 |  **82.59±0.98** | 52.79±0.29 | 72.10±0.66|
> | GraphRAG |75.24±0.50| **81.57±0.94**  | **77.48±0.76** | 80.70±0.90 | 52.42±0.27 | 72.50±0.68|
> | HippoRAG | 70.48±0.44| 80.18±0.87| 74.32±0.72 |  81.65±0.95 |  **56.13±0.35** | 72.64±0.67|
> | RAPTOR  |  **76.67±0.53**   | 80.65±0.93 | **77.48±0.78**|  82.28±0.97 | 54.83±0.32 | **73.58±0.71**|
>
>
> > Response to suggestions and questions
>
> S1. Motivation
>
> The existing datasets are mostly based on wiki, medical, or inquire about various movies and celebrities. LLM has already extensively encountered these in the training process. Therefore, we chose the more challenging domains that have certain thresholds and are less frequently encountered by LLM. At the same time, in practical application scenarios, GraphRAG is more often utilized in specific fields to construct precise, professional, and efficient graphs, rather than building an extremely large and resource-consuming universal knowledge graph.
>
> S2. Reviewer Report.
>
> Thanks for the suggestion. However, the auto generated reviewer report is invisable to authors. Would you please provide the problems indicated in the report? Thanks!
>
> Q1. Experts' info.
>
> A total of 8 experts participated. They are researchers holding a doctoral or master's degree in computer science and have conducted in-depth research in this field.

---

> > ### Comment · Reviewer_FHaj · 2025-08-09
> >
> > W1. Thank you for pointing out the difference in my previous comment of works that were missed in being cited and those lacking detailed comparison for the novelty of the proposed framework. The inclusion of additional experiments is appreciated, although more discussion on what specific novelty is being brought beyond the newly collected dataset of computer science questions is still lacking.
> >
> > W2. Regarding the evaluations, many works are developed with a specific focus, such as improved token efficiency that might come with a trade-off in other aspects, so the separate evaluations are indeed needed. Also, I agree that exact matching for open-eded questions is not appropriate. However, I still believe the use of LLM-as-judge for a benchmark is less appropriate as compared to ground-truth answers, e.g., for fill-in-the-blank, where it is likely still possible to avoid the LLM evaluation.
> >
> > W4. Thank you for clarifying your usage of “reasoning” which makes more sense now.
> >
> > W5. Just to be clear here, the concern was not specifically focused on randomness in the result, but was the lack of sound statistical evaluation of the results. At a minimum the +- should be listed if not going to conduct further statistical tests. Also, listing the response as “No” as compared to “N/A” would have at least exhibited knowledge that this information should be included although not done so in the first version of the paper (i.e., including second-order statistics for results is practically required empirical result disclosure at this point). With that said, partially including them here is appreciated.
> >
> > W/S.1. I’m not sure I’m sold on the fact that existing datasets for medical, movies, etc. topics are already covered in LLM training and therefore not good benchmarks, but computer science textbooks (that are also almost surely included in LLM training) is more appropriate. However, I’m not a domain expert to determine the difficulty of questions across datasets. That said, the performance differences across methods here appear to be less significant, e.g., reasoning ability averages on GraphRAG-Bench for various methods (across all question types) were 59.43, 60.17, 60.46, 60.36, 60.90, 60.81, with a +- around 0.8 would suggest that this may actually not be such a good benchmark if all methods are performing near identical to each other (on average). (Similar with other accuracy related evaluations).
> >
> > W/S.2. It mentioned that 2 files were inaccessible. However, upon further investigation I am actually not sure why the checklist is listing this without details on what is missing and unable to figure that out (but did not download and run all the data/code myself to verify this). It does also mention the Responsible AI Fields are all missing, e.g., rai:dataBiases.
> >
> > W/Q.1 There seems to be inconsistency in what you’re describing as the “experts” in your response that was not discussed in the paper. In one place you mention “researchers with a compute science doctoral degree” and another you mention “they are researchers holding a doctoral or master’s degree in compute science” which further signals the importance of having these detailed included in the paper. Thank you for conducting and including the reliability analysis on the experts. However, concerns still exist on the expert evaluators, if this were a research track paper I would not have such concerns, but given the lack of details even after asking the first time makes me believe this would benefit from more time to develop those details and sound evaluation.
> >
> > Additionally, for a potential camera-ready version of this work it would help to clarify the difference between the two recent papers both proposing “GraphRAG-Bench”:
> > 1. GraphRAG-Bench: Challenging Domain-Specific Reasoning for Evaluating Graph Retrieval-Augmented Generation
> > https://arxiv.org/abs/2506.02404
> > 2. When to use Graphs in RAG: A Comprehensive Analysis for Graph Retrieval-Augmented Generation
> > https://arxiv.org/abs/2506.05690
> >
> > In summary, after reading your response, and the other less detailed/critical reviews, although I’m mostly inclined to keep my original score due to my continued concerns, I believe this benchmark may still have a positive impact to the research community, and therefore will increase 3->4.

---

> > > ### Author Response · Authors · 2025-08-09
> > >
> > > Dear Reviewer FHaj,
> > >
> > > We sincerely appreciate your detailed guidance and serious comments in response to our rebuttal. It is a great privilege to be acknowledged by an expert like you, and also inspiring given your increased score. We are also impressed by your comments again, which will greatly help us to improve the quality of the manuscript, and we are happy to have more discussions with you regarding the new points.
> > >
> > > > W1. Specific novelty brought by CS.
> > >
> > > Thanks for your careful check. We will add the advantages of using CS and AI in the final version to showcase our unique contributions. We highly value this domain for three reasons. **First**, CS and AI naturally contain a variety of types of challenging tasks compared with all the existing 'single-type' datasets, where the questions are extremely hard which requiring a college-level command of knowledge, compared with existing commonsense datasets. In CS and AI domain, we have `conceptual understanding` tasks (algorithms, networks, databases, paradigms, machine learning, NLP, CV), `Math problems` (Discrete mathematics, linear algebra, matrix computation, calculus), `Code generation` problems (programming, optimization, code completion, error detection), etc. **Second**, the performance on LLMs on CS and AI is still very unsatisfactory or even bad. It is valuable to benchmark and research. **Third**, it remains a blank in the benchmark of LLMs considering CS and AI. This also brings two opportunities. (i) LLMs will have few chances to be fine-tuned based on our questions. (ii) Following research on GraphRAG or LLMs themselves could benefit a lot from our benchmark.
> > >
> > > >W2. Evaluation of fill-in-blank questions.
> > >
> > > We believe that exact matching sometimes could misjudge the actual correct content (such as different expressions of nouns, the/a/an and other words). After our post-evaluation, the results of the fill-in-blank judgment assisted by LLM are more accurate and suitable for this kind of open-ended question.
> > >
> > > >W/S.1. Marginal performance differences.
> > >
> > > Our observations about the marginal performance differences align with yours indeed. Particularly, we have investigated the overlap among different methods for the correctly answered questions. The results reveal that these seemingly similar aggregate scores mask important observations in how different methods excel at distinct question types and difficulty levelsWhile RAPTOR is a hierarchical method, it is good at global summarization and cross-domain questions, GFM-RAG is good at multi-hop reasoning which forms a chain-like path.
> > >
> > > We will add the statistics in the final version.
> > > | Hops | Questions | Percentage | Difficulty | Example Type | Evaluation Focus |
> > > |------|-----------|------------|------------|--------------|------------------|
> > > | 1    | 128       | 12.6%      | Easy       | Single-fact retrieval | Basic factual accuracy |
> > > | 2    | 342       | 33.6%      | Medium     | Dual-condition reasoning | Simple logical combination |
> > > | 3    | 287       | 28.2%      | Hard       | Cross-entity relational reasoning | Multi-step reasoning coherence |
> > > | 4    | 178       | 17.5%      | Very Hard  | Multi-domain knowledge fusion | Long-range dependency handling |
> > > | 5    | 83        | 8.1%       | Expert     | Counterfactual reasoning with implicit constraints | Complex logic & noise resistance |
> > >
> > > >W/S.2. All results are reproducible based on the code and data we have published. We will include explicit links in the final version.
> > >
> > > >W/Q.1 Thanks for your careful check. We will clarify this in the paper that the experts involved in the construction of the dataset are eight experts with a doctoral or a master's degree in computer science, and the LLM alignment is performed by the majority voting among three of them with a doctoral degree in computer science. Thanks for your reminding, we will explain these further in the latest version.
> > >
> > > In general, thank you so much again for your suggestions and comments. We will carefully revise the manuscript according to all your suggestions.

---

### Decision · Program_Chairs · 2025-09-18

**Decision:**

Reject

**Comment:**

The article introduces a benchmark for GraphRAG systems that evaluates multi-step reasoning, retrieval, and cost metrics, addressing the lack of standardized evaluation tools in this area. The benchmark is curated, well-documented, and aligned with the track's goals. Reviewers agreed on the strength of its design and the potential to stimulate research in this growing subfield. The work fills a gap in evaluation infrastructure and provides a useful resource.

Concerns were raised about ethical issues, and two ethics reviewers suggested improvements; the authors have responded positively. During rebuttal, the authors clarified licensing, provided safeguards, and argued for extensibility. Reviewers maintained their accept ratings, and some increased them. Given the timeliness of the topic and the lack of similar benchmarks, the article is a strong submission; we support its acceptance.

===== FINAL UPDATE FROM DB Track PCs ====

The final decision for this paper has been taken by the program chairs after consultation with the SACs. All Senior Area Chairs have ranked papers according to the feedback from the AC during the review process. We decided to leave the original meta-review to reflect the opinion of the AC in light of the initial discussions with reviewers and SAC.